# Back to the Origins: Background and Perspectives of Grapevine Domestication

**DOI:** 10.3390/ijms22094518

**Published:** 2021-04-26

**Authors:** Fabrizio Grassi, Gabriella De Lorenzis

**Affiliations:** 1Department of Biology, University of Bari, via Orabona 4, 70125 Bari, Italy; 2Department of Agricultural and Environmental Sciences, University of Milan, via G. Celoria 2, 20133 Milano, Italy; gabriella.delorenzis@unimi.it

**Keywords:** crop wild relatives, demography, grapes, introgression, palaeogenomic, pan-genome, selection, *sylvestris*, *Vitis*, wild

## Abstract

Domestication is a process of selection driven by humans, transforming wild progenitors into domesticated crops. The grapevine (*Vitis vinifera* L.), besides being one of the most extensively cultivated fruit trees in the world, is also a fascinating subject for evolutionary studies. The domestication process started in the Near East and the varieties obtained were successively spread and cultivated in different areas. Whether the domestication occurred only once, or whether successive domestication events occurred independently, is a highly debated mystery. Moreover, introgression events, breeding and intense trade in the Mediterranean basin have followed, in the last thousands of years, obfuscating the genetic relationships. Although a succession of studies has been carried out to explore grapevine origin and different evolution models are proposed, an overview of the topic remains pending. We review here the findings obtained in the main phylogenetic and genomic studies proposed in the last two decades, to clarify the fundamental questions regarding where, when and how many times grapevine domestication took place. Finally, we argue that the realization of the pan-genome of grapes could be a useful resource to discover and track the changes which have occurred in the genomes and to improve our understanding about the domestication.

## 1. Introduction

*Vitis vinifera* L. is one of the most economically important fruit crops in the world. It is used in the global wine industry, covering approximately 7.5 million hectares in 2019 and producing more than 67 million tons of grapes [1]. Two subspecies are recognised: the wild form, *V. vinifera* subsp. *sylvestris*, and the domesticated one, *V. vinifera* subsp. *vinifera* (or *sativa*). Subspecies *sylvestris* is considered the progenitor of the subspecies *vinifera* and, phenotypically, the two subspecies differ in some traits relating to the morphology of flower, seed and leaf, to berry and bunch size (Figure 1) and sugar content [2]. The flowers of *sylvestris* individuals are dioecious (male and female flowers), while those of the *vinifera* are monoicous (hermaphrodite flowers) [3]. A recent study proposes that dioecy was lost during the domestication process through a rare recombination event which occurred during sexual reproduction [4]. The subspecies *sylvestris* was neglected for several years, but recent studies have evidenced a great morphological variability and a presumed subdivision in different variants [5,6]. Moreover, ecological studies have shown that the wild populations, differently than *vinifera*, can grow in a wide range of habitats with wide adaptability to different soils, including forested wetlands, along seasonal rivers in closed forests and sand dune shrublands [6].

The grapevine domestication process occurred in the Near East, in some place along a large area that ranges between Central Asia and the Black Sea [7]. An area satisfying the features of a primary domestication centre, such as durable interest in developing viticulture, an area with a high population density, stable settlements and at the crossroads of trade flows and cultural trends. Archaeological and archaeobotanical evidence suggest that viniculture began in the Near East, ca. 6000–5800 BC during the early Neolithic Period [8]. However, some questions about where and when the process of domestication began, remain elusive. Evidence of winemaking is based on traces of tartaric acid from pottery residues also commonly found in other fruits [9]. Moreover, the recognition of the two forms of grapevine (*sylvestris* and *vinifera*), based only on fossil seeds, is tricky and the morphological change of grape pips is indicative of consolidated winemaking practice but not informative about the beginning of domestication [9].

The current dispersal area of wild subspecies ranges from Western Europe to Western Himalaya, in areas 900–1000 m above sea level [10]. To date, the subsp. *vinifera* is cultivated in Europe, Asia, Northern America, Southern America, Northern Africa, South Africa, New Zealand and Australia. The domesticated form includes a huge number of varieties, with more than 10,000 cultivars believed to exist in the world [11], complicating the identification of cultivar and the tracking of the species origins. The huge genetic variability available nowadays can be associated with breeding, vegetative propagation and mutations during the evolution of grapes. Due to the growing socio-economic impact of the wine sector around the world, there has been an increase of works studying the grapevine genetic resources and investigating its origin. Ampelography, based on phenotypic observations, was the first method which scientists had to investigate the genetic diversity and origin of grapevine genotypes. Although ampelography has shown, over the years, a whole series of negative aspects, one among many being the influence of the environment on the expression of phenotype, Negrul [12] classified the wild and domesticated varieties in *Proles* based on the phenotype. Individuals from Central Asia and Caspian Sea were included in the *Proles orientalis*, the ones from Central and Western Europe in *Proles occidentalis*, the ones from Eastern Europe, Georgia and Turkey in *Proles pontica*. *P. occidentalis* individuals have small berries, high acidity and scarce sugar content. *P. orientalis* individuals have opposite traits, while *P. pontica* individuals show intermediate traits, such as scarce acidity and discrete sugar level.

Over the years, in terms of domestication, the necessity to move from a detailed to an overall view arose. For this reason, initially, the researchers included a pool of samples coming from a limited single area of interest, then they involved an increasing number of samples coming from different wine-growing areas, covering large geographic regions [13,14,15]. This shift was also made possible thanks to the revolution in molecular investigation techniques. In the last two decades, molecular studies have led to a deeper knowledge of the evolution of grapevines and the domestication process continues to be at the centre of debate in several studies [16]. In this review, we are going to summarise and discuss the principal phylogenetic and genomic studies, trying to clarify basic queries concerning when, where and how many times grapevine domestication took place.

## 2. What Is Lurking behind the Domestication Process?

The study of domestication is a research area that does not only concern how plants were modified by humans in the past but allows us to understand how to design ideal crops for more sustainable agriculture [17]. This aspect is essential because the genetic diversity of crops has been reduced throughout the process of domestication resulting in the loss of several traits [18,19]. Moreover, breeding processes that have involved the selection of desirable traits to improve the crop productivity, have aggravated the situation, causing the loss of disease-resistant traits. The depletion of diversity has a serious impact on agriculture, constraining the possibility of the cultivation of crops in more extreme environments or in alternatively increasing the input of pesticides and water, with severe cost to the environment [20]. To reverse this situation, it is necessary to identify the wild relatives of crops which are known to tolerate biotic and abiotic stress, a trait lost during human selection.

Generally, domestication occurred in areas of high biodiversity and some authors have proposed that over 160 taxonomic families and 2500 species could have been involved [21,22]. Plant domestication is recognised as a geographically restricted evolution process, by which a crop originating from a wild progenitor was successively spread and cultivated in outermost regions [23]. According to Vavilov’s hypothesis [24], domestication occurred in the centres of diversity of respective species but recently there has been considerable debate which opposes a single centre of origin versus multiple origins. Crop species have been influenced by human activity over thousands of years and signatures of the domestication process have been deeply impressed in the genomes [18,25].

If compared with the evolution of species, several changes may have been collected in the domesticated form throughout time, forming an independent monophyletic group. However, unlike speciation in which we can delimit the split between two species at the end of the process, the domestication process can potentially be considered an infinite process. At any time, the human-caused reunion between crops and its wild relatives can trigger the formation of gene flow with the consequent development of introgression events, while an increase of divergent traits can determine reproductive isolation between different forms so that they are no longer able to produce fertile progeny. Moreover, the domestication can revert at any time through a feralisation process in which the crops return to the wild environment.

Usually, crops show evidence of genetic bottlenecks that remain well imprinted during initial domestication, but the intensity can vary from crop to crop and additional bottlenecks or gene flow may occur confounding the reconstruction of the domestication process [23]. This can be particularly true for perennial species such as the grapevine, in which the intense human activity of cultivation, breeding and trade in the Mediterranean basin, which has occurred for several thousands of years, may have tangled the evolutionary relationship. Moreover, it is more likely that grapevine domestication was not a single step process but rather a multistage progression. As for other crops the process would have included a pre-domestication period when humans planted and cultivated wild plants, a diversification period with an increase in the frequency of desirable traits, an adaptation period to different agro-ecological environments and finally a breeding period to increase yield and quality [26]. To define when the different phases began is particularly complex, but, in recent years, some studies based on modern molecular methodologies have endeavoured to clarify these aspects.

## 3. Where and How Many Times Has Grapevine Domestication Taken Place?

Whether the grapevine was domesticated only once, or whether some varieties were domesticated independently, is a mystery hotly debated and different scenarios are proposed (Figure 2). The main hypothesis defined as the “Noah hypothesis”, so named in honour of the biblical patriarch who planted the first vineyard on Mount Ararat after the flood, proposes that grapevine domestication processes took place in a well-defined restricted area (Single-origin model). In addition, a multiple-origin hypothesis that provides for the foundation of independent lineages originating from wild progenitors spread some place along the entire distribution range has been proposed (Multi-origin model) [27,28].

Based on the anthropological condition, historical and ampelographic evidence, additional grapevine domestication centres in Europe were hypothesised [29]. A secondary grapevine domestication centre emerged in the Middle Bronze Age in the Greek region closest to the Caucasus, where the contributions of the nearby primary domestication centre gave the decisive push to the transition from embryonic viticulture to domestication. This process was repeated by the Greek colonisation of Sicily and Southern Italy, starting from the Iron Age (tertiary domestication centre), and during the Punic, Greek and then Roman colonisation in South-Eastern Iberia, where a quaternary centre of domestication emerged. Later, a fifth domestication centre was established in Northern Italy, Southern France and North-Eastern Spain [29]. Following this idea, the main grapevine migration routes around the Mediterranean basin have been proposed: (*i*) from Mount Ararat to Greece through Mesopotamia and Egypt or through Anatolia; (*ii*) from Greece to *Magna Graecia* (Sicily, Southern Italy), France (Marseille) and Spain; (*iii*) from France to the north of Europe [30,31] (Figure 3).

Genetic relationships between wild and domesticated forms can be traced by molecular analysis and the microsatellites are widely used to show the genetic structure and domestication history of crops. In the early 2000s, Sefc et al. [32] showed that it was possible to determine the geographical origin of grapevine by analysis of their genotypes using a set of nine microsatellites. The markers used were found to be variable and informative, and thus specific assignment tests were applied to estimate the likelihood that each accession belongs to a given area of origin. These results have opened new possibilities to explore the origin of grapevine and to test specific hypotheses that were previously suggested.

The first large-scale genetic characterisation study to explore the origin of 244 grapevines, including also wild accessions was carried out by Aradhya et al. [33]. The genetic variability, investigated by multivariate and cluster analysis, has confirmed Negrul’s distribution and classification that supports three groups spread progressively from the Near East to Europe. Moreover, Aradhya et al. [33] evidenced a close affinity between domesticated cultivar and the wild progenitor in south-western France suggesting the local origin of some old grapevine. In the following years, numerous population genetic studies, based on an increasing number of nuclear microsatellites, were published. The main aims were to investigate the genetic structure of germplasm resources [34,35,36], to explain the relationship between *vinifera* and *sylvestris* [37,38] and to establish the parentages of relevant cultivars [39,40,41,42,43]. A preponderant divergence between *vinifera* and *sylvestris* has been observed by several authors [38,44,45] but some studies have also identified a genetic signal that suggests the contribution of wild plants in the domestication process which occurred outside the main domestication area [37,46,47,48]. Studies carried out on germplasm resources collected in the Caucasus have shown an unexpected diversity and richness [49] raising some doubts about the correct geographic place of the main domestication area [35]. The results of two large-scale genetic studies, based on the variability of nuclear microsatellites analysed by Bayesian approaches, have evidenced that the identification of the main centre of domestication is more complex than thought and probably extended into many Central Asian countries (Figure 3) [15,50]. While Riaz et al. [15] have suggested Georgia as an ancient centre of grapevine domestication and evidenced the involvement of Western Europe germplasm in a second centre of domestication, Bacilieri et al. [50] have proposed that the Iberian and Italian Peninsulas are regions of mixing and exchange of varieties. This last finding has been interpreted as the result of an intense activity of exchange by Romans through the combined action of selection, breeding and migration [50].

Although microsatellites are widely recognised as an important tool to provide insights about the recent phases of breeding, they are less so when ancient events of domestication are explored. Moreover, several authors have observed severe limits that can affect the results. First, some studies have proposed that microsatellites may suffer from homoplasy [51]. Microsatellite alleles are generally revealed by electrophoretic methods and fragments identical in size are not necessarily identical by descent due to convergent mutations that occurred in the lineages. This issue is often unexplored but empirical data have demonstrated that the population structure can be affected especially when genetic markers with high variability are used [52,53,54]. Secondly, variation in flanking regions due to genetic divergence between grapevine subspecies may trigger the occurrence of null alleles, introducing important biases into genetic studies [55]. For example, the presence of null alleles produces an overestimation of the homozygotes causing deviations from Hardy-Weinberg equilibrium expectations and the populations affected can show an artificially reduced variability [56].

One of the modern challenges in plant science is to improve the access to and use of genetic variability hidden in the genomes. In the genomic era, efficient genotyping tools should be able to cover a large part of the genome. For this reason, Myles et al. [57] have examined over 70,000 single nucleotide polymorphisms (SNPs) discovered by Illumina GA sequencing, and then a Vitis9kSNP array that includes polymorphisms from different grapes has been planned. Myles et al. [58] have observed a reduction of genetic variability, confirming that the grapevine has suffered from a weak domestication bottleneck in the Near East followed by diffusion towards Europe. Intense signals of reduction in genetic variation are common in annual crop species, while perennial crops seem to have suffered a relatively mild genetic bottleneck. Miller and Gross [59] reported that several factors might have contributed to the relatively mild genetic bottleneck in perennial species such as multiple origins of lineages, somatic variations collected during vegetative propagation, outcrossing by sexual reproduction and gene flow between domesticated and wild forms. Moreover, in grapevine other reasons could be an insufficient sampling of *sylvestri*s or the habitat destruction and consequently extinction of populations [60,61].

In the last few years, new arrays holding tens of thousands of SNP loci have become available, increasing the capacity to investigate the genetic variability hidden in the genomes. Marrano et al. [62] array includes around 37K SNPs, identified in a group of 51 *vinifera* and 44 *sylvestris* genotypes, while Laucou et al. [14] array, called Vitis18kSNP genotyping array (Illumina), includes around 18K SNP identified in a group of 47 *vinifera* and 18 non-*vinifera* genotypes. Due to the high-throughput of genotyping arrays, these tools have been used mainly to investigate the genetic variability of large grapevine collections, such as the Vassal repository (France), IMIDRA and ICVV repositories (Spain), JKI repository (Germany) [14] and some Italian collections [63,64,65,66,67]. SNPs profiles have revealed a high genetic diversity in accessions collected in the southern regions of Caucasus and a gene flow from East to West that confirms the hypothesis of a main domestication centre located in Near East [13,63,66]. Moreover, due to the common genetic background between Southern Italian and Greek grapevine, it has been proposed that Southern Italy has played the role of a bridge between Greece and Central Europe (Figure 3) [66]. These results can be better understood taking into account the migration routes of Neolithic populations. Three migration routes could have been taken by Neolithic farmers to reach Europe, two by sea and one by land. The first sea route was from the Aegean Anatolian coast to Mediterranean islands and Southern Europe [68], the second was from the Levantine coast to Aegean islands and Greece [69], while the overland route was from North-Eastern Anatolia to Thrace and the Balkans [68]. Paschou et al. [70] demonstrated that European colonisation took place mainly by sea, via Crete, and only secondarily by land. Furthermore, Northern Italy has been found to be an admixed centre, where Southern Italian and Central European populations converged (Figure 3) [66]. However, further studies are needed to understand the contributions made by the wild populations located in the Italian and Iberian Peninsulas, France and Greece (including the main islands of the Mediterranean basin [42,44]) and to understand whether these locations have had a role of secondary domestication or were diversification centres.

## 4. Plastid DNA to Explore the Maternal Lineages

Although plastid genomes tell one side of the evolutionary history due to the uniparental inheritance, molecular markers are very helpful for exploring the origin of several plants [71]. Preliminary analysis of plastid simple sequence repeats carried out on grapevine accessions have shown that the genetic variability reflects the geographic distribution, suggesting these molecular markers might be able to resolve the intricate puzzle of the origin of grapevine [72,73]. Plastid microsatellites are successively used to explore the haplotype diversity from grapevine cultivars distributed from the Near East to the Mediterranean basin. Even though conclusions cannot be drawn about the domestication area from the results, the authors have evidenced the existence in the past of an intensive varietal exchange of germplasm and propagation throughout Europe [74,75]. On the other hand, Arroyo-García et al. [76], have examined the haplotype relationships under a network model in a large sampling of wild and domesticated accessions. The results supported the existence of at least two centres of domestication, one in the Near East and another in the western Mediterranean region, confirming the involvement of several founders recruited throughout a prolonged time period. Similar findings have also been made by Cunha et al. [77] that, analysing grapevines collected from the Iberian Peninsula, have reinforced the hypothesis of a secondary domestication centre located in Western Europe.

In phylogeographic studies, findings about origin, conservation and diversity of grapevine are obtained analysing the haplotypes distributed in wild populations collected from different geographical areas. The Caucasus, harbouring the highest haplotype diversity observed in plastid genome, was proposed as the centre of origin of the species, while the Iberian and Italian peninsulas are the result of refugial persistence and accumulation of variation over several ice ages [61]. High levels of haplotype variability in wild grapevine populations collected from different geographic zones of the South Caucasus regions have been confirmed in successive studies [78]. Moreover, phylogenetic analysis showed that wild grapevine samples collected in Caucasus constitute the oldest lineage among those examined [79], supporting the hypothesis that the Mediterranean lineages diverged successively [80].

Unfortunately, the exploration of plastid DNA has only partially resolved the evolutionary origin and domestication of grapevine, probably due to the low number of loci analysed or the slow mutation rate of grape plastomes [81]. Moreover, incomplete lineage sorting [81] or introgression [73] could have confuse phylogeny between close relatives. An increasing number of plastid DNA sequences were used to explain the evolution of some *Vitis* taxa [79,82,83,84] as well as to support the reconstruction of the pedigree of some cultivars [85]. Today, next-generation sequencing technologies offer an unprecedented access to complete genome sequences and to investigate the plastomes has been proven to be a good procedure to infer new evolutive lines [86,87]. For example, maternal lineages could be explored with the aim of identifying the times wild lineages were introduced to the Mediterranean basin, the main areas of domestication and successive routes of diffusion. Moreover, there is little knowledge about the phylogenetic relationships between *V*. *vinifera* and other grapes. Some authors have proposed the origin of an ancient Eurasian clade [79,85] and *V. jacquemontii* R. Parker could be the sister species of *V. vinifera* [88]. The identification and characterisation of closely related wild grapes can have important repercussions also from an agronomic perspective. Thus, the analysis of plastomes combined with adequate sampling could make it possible to test specific hypotheses and lead to new and fascinating conclusions.

## 5. When Did Grapevine Domestication Occur?

Domestication is defined as a process by which a species becomes adapted to human use by selecting desirable traits occurring over generations. In this way, one or more populations are separated from their natural ecological context and placed under anthropogenic-driven pressures [89]. Generally, it has been believed that the domestication process was a rapid phenomenon [90] which started recently (<12,000 years ago), because theoretically, few cycles of selection are needed to split domesticated varieties from wild ancestors [90]. In the last few years, archaeological and genomic findings have suggested that the pre-domestication phase may have lasted several thousands of years and that the relationship between humans and crops could have persisted longer than thought [91,92].

Demographic inference methods are available today to analyse the population size changes overtime and genome-wide resequencing methods are applied to infer the population structure and demographic history [93]. Zhou et al. [94] have evidenced that wild and domesticated grapevines diverged from 22,000 to 30,000 years ago experiencing a continual reduction of population size and that only recently table grapes have been separated from wine grapes (2500–2600 years ago). The long decline observed could be the result of a long period of pre-domestication management that began before archaeological evidence. In support of this claim, the authors note that other crops [95] have experienced a protracted period of population size reduction and that Southern Caucasus regions have evidence of human activity for more than 20,000 years. However, Zhou et al. [94] have used a rate of mutation of 2.5 × 10^−9^ per nucleotide per year previously observed in the *Brassicaceae* family [96]. Mutation rate is a major source of uncertainty in demographic analyses based on genomic data and heterogeneity in the rate was often observed between plants that show biological and lifestyle differences (e.g., woody vs. herbaceous) [97,98]. Thus, it is reasonable to believe that perennial crops that show intensive vegetative propagation could collect mutations at a slower rate, showing a shift in the time frames of demographic history. Recently, Liang et al. [99] have proposed a new scenario in which the time of divergence between wild and domesticated grapevines was 200–400 Kya. These new conclusions greatly predate the start of the domestication process. We underline that differently from Zhou et al. [94], Liang et al. [99] have decided to estimate the mutation rate, proposing 5.4 × 10^−9^ mutations per nucleotide per year in grapevine. Although these studies agree with the recent theories that propose a long pre-domestication phase for the crops [91,100], some issues remain pending and the findings should be interpreted with caution. For example, the choice of mutation rate is a fundamental step in demographic analysis and, indeed, different estimation methods can produce different results. Moreover, we underline that the conclusions about divergence times between lineages are circumscribed to the plants sampled that represent a subset of proto domesticated grapevine. For example, some populations could have originated from a restricted group of individuals imported by humans for breeding activity or have suffered introgression from other taxa [101]. The inclusion of these populations in the demographic analysis could influence the results. Thus, we highlight that new insights could be obtained if samples from distinct geographic areas are included in future analysis.

## 6. Introgression between Wild and Domesticated Grapevine

Hybridisation is a natural process generally attributed to taxa range changes but in certain circumstances can be imputed to human activity. In recent years, molecular studies have proved that interspecific hybridisation and introgression are central processes in the evolution of *Vitis* genus [79,82,102,103]. Introgression is considered a process mediated by the transfer of genes through repeated backcrosses and when occurring from crops to its wild relatives can have deleterious consequences on the genetic structure and conservation of populations [104]. On the other hand, the inverse process of introgression from wild to crop may be considered a rapid method for the crop to adapt to new environments (Figure 2). The admixture between perennial crops and wild progenitors has hardly been studied and genetic analysis was not always able to trace the gene flow, especially if an elevated number of generations have elapsed.

Some authors have questioned the existence of a really wild grapevine suggesting that wild plants observed in nature can be feral [2,105,106]. Admixture in grapevine is strongly supported by the diffusion of rootstocks and cultivars abandoned and then naturalized in the wild environment and by the fact that the *sylvestris* and *vinifera* have proven reproductively compatible [107,108]. Although most of the genomes of the two subspecies are shared, some studies have demonstrated that the allele frequencies observed between them are divergent enough to evidence the origin of each individual. Grassi et al. [37] have shown that Lambrusco varieties have a genetic intermediate position between *sylvestris* and *vinifera*. Zecca et al. [106], using different Bayesian approaches, showed that 10% of individuals sampled in Sardinia had experienced a cryptic introgression and that backcrosses to wild grapevine occur more frequently than backcrosses to cultivars. De Andrés et al. [44], using parentage analysis identified that 19% of the analysed genotypes are derived from crosses between wild and cultivated grapes in Spain. Myles et al. [57], analysing a set of SNPs with a specific 3-population test for admixture, have proposed a scenario in which European cultivated grapevines have suffered introgression from *sylvestris*. D’Onofrio [109] showed that the 9% of supposed wild individuals are indeed spontaneous crosses with most widespread cultivar in Tuscany and underlines that wild germplasm may have contributed to produce some cultivated varieties. Arnold et al. [110], analysing large populations located in the Donau-Auen National Park (Austria), evidenced that 8% of plants were hybrids and proposed that these plants may belong to a more complex taxa that also involve different grapes, used in the past as rootstock and neglected to date.

Generally, in genetic studies a preliminary selection based on morphological and ecological traits is conducted to exclude feral in successive analysis thus the real introgression in nature might be particularly intense and involve far more than 10–15% of the samples. Today little is known about the effect of introgression, but the continuing extinction of *sylvestris* plants with a significant reduction in gene diversity and heterozygosity of populations in respect to *vinifera*, is causing a serious demographic decline of wild grapevine [44,108,111]. An increasing number of studies have proposed that the maintenance and the conservation of wild populations should be a primary aim in Europe because these plants are a unique and fundamental genetic resource for the improvement of the cultivated grapevine in the future [45,60,80,112,113,114,115]. Some molecular studies have been proposed recently with the aim of studying genomic divergence between the two subspecies [45,62,116], but greater efforts should be made to explore the real level of introgression. Moreover, some urgent issues which are still pending should be examined such as the effective direction and intensity of gene flow between wild and domesticated forms as well as the accurate estimation of admixture times (Figure 2). Indeed, both models that involve introgression presented in this study (Figure 2C,D) might have occurred in the past. Future works should be directed to identify which genomic regions are involved in gene flow. For example, signals of introgression might not be diffused in the entire genome but localised in specific chromosomes.

The increasing accessibility to genomic data and the new tools of evolutionary genomics developed recently are improving the ability to trace the process of admixture. For example, different approaches that integrate population genomics and phylogenetic methods could be used to co-estimate both splits between subspecies and migration events. The TreeMix model [117], developed to address historical relationships, showed a good efficiency when employed to resolve the evolution of crops and their wild relatives [118,119], but also unravelled the intricate relationship between grape hybrids [81]. Furthermore, the D-statistics analysis (ABBA-BABA tests) have already been applied to detect introgression among sympatric populations of Asian grapes providing interesting findings [120] while inferential tools for historical demography such as Approximate Bayesian Computation (ABC) methods could be used to estimate different demographic parameters such as divergence times and migration rates.

## 7. Ancient DNA to Investigate Grapevine Domestication

A limit of the research aimed to investigate the origin of grapevine domestication is, undoubtedly, the interpretation of the past through the study of modern cultivars. An important contribution in dealing with this issue comes from archaeological remains. So far, the most common method used to study the diversity of archaeological pips has been based on morphometric analysis. Wild and domesticated grapevine seeds can be distinguished based on seed shape, contributing greatly to connecting past to present diversity [121,122,123]. Basically, *sylvestri*s seeds are spherical with a small beak, while the *vinifera* seeds are pyriform-shaped with a well-developed beak [124,125,126]. Thanks to morphological analysis of archaeological seeds, new insight on secondary domestication centre has been suggested, such as a domestication centres in the Languedoc region in France [124], and an early local grapevine domestication in prehistoric Greece [125]. The combination of seed morphological and genetic characterisation should allow a better understanding of the grapevine domestication process. Nevertheless, if the genetic characterisation of modern varieties had a great impact among researchers with very successful results, the same could not be said about the study of ancient DNA (aDNA). The first results on genetic diversity of grapevine aDNA were not completely satisfactory [127,128,129,130,131,132] for several reasons. First, aDNA from archaeobotanical specimens must be correctly preserved and only charring and waterlogging samples are a good source for DNA amplification [127]. Secondly, aDNA should not be contaminated by exogenous DNA Thirdly, a high amount of template DNA is needed [133].

The next-generation sequencing techniques provided a significant boost to aDNA studies. Indeed, short read length makes this new technology ideal for aDNA research, guaranteeing reliable results [134]. Wales et al. [135], for the first time, have high-throughput sequenced aDNA from archaeological grape specimens coming from Armenia, England, Israel, Italy and Turkey, dating ca. 4000 BCE–1500 CE. Both the plastome and nuclear genome were sequenced. Plastome analysis showed evidence of genetic introgression from *sylvestris* to *vinifera* in Western Europe, as hypothesised by Arroyo-García et al. [76] and Myles et al. [58]. Although the number of enriched nuclear loci was limited, nuclear data were more informative than the plastome ones. The comparison of nuclear aDNA data against a reference panel of domesticated and wild genotypes highlighted a strict correlation between archaeological specimens and Eastern or Western European grapevines.

The first study on nuclear aDNA was published in 2019 by Ramos-Madrigal et al. [136]. The authors presented data of targeted enrichment and shotgun sequencing of around 10K SNP loci from French archaeological pips dating from the Iron Age to Medieval period. The SNP loci were selected from the Vitis18kSNP genotyping array [14], so that it was possible to compare the SNP polymorphisms of archaeological specimens with the ones of a reference panel providing data for domesticated and wild genotypes, in order to identify relationships between ancient and modern varieties. This comparison led to the conclusion that archaeological seeds probably originating from domesticated grapevine individuals, supporting Bouby et al. [121] hypothesis that many pips from Roman and Medieval period originated from domesticated grapevine, even though their shape looks like that of wild individuals.

A further step forward towards understanding the grapevine domestication process has been taken by Bouby et al. [122], when geometric morphometric and palaeogenomic investigations on grape pips from the Georgia (Caucasus), covering a period ranging from Neolithic to the Roman period, were combined. Based on the morphotype, most of the pips showed a seed morphology similar to the modern cultivars from Caucasus, Southwestern Asia and Balkans, suggesting a very long-standing connection between modern and ancient diversity. Similar results were obtained by the comparison of aDNA SNP profiles with the modern SNP data, although both ancient (three) and modern (around 20 Georgian cultivars) data do not cover the entire past and present genetic diversity in Georgia. Taking into account these challenging results, further studies combining morphometric analysis and aDNA data on a large group of archaeological samples collected in the first grapevine domestication centre and along the main migration routes are expected.

## 8. From Genome to Super-Pangenome

All genetic variations, that ranging from SNPs to large genomic structural variants (SVs) such as duplications, inversions and transpositions, are the result of selection through time. The origin of SVs in plants is relatively poorly understood as are the mechanisms that govern the gene gain and loss, even though many important agronomic traits may be determined by these changes [137], such as grapevine berry colour and bunch shape determined by insertion of Gret1 [138] and Hatvine1-rrm [139] transposable elements, respectively. Due to domestication, the grapevine has experienced a bottleneck [58] and vegetative reproduction led to the accumulation of recessive deleterious mutations [94]. On the other hand, the stability of phenotype has made the clonal propagation attractive in agriculture [21]. Grapevine is an ideal candidate to study genetic variations in clonal systems and, over the last few years, many resequencing studies focus on the effects of SVs throughout domestication. Zhou et al. [140] showed that domesticated grapevine, compared with its wild dioecious progenitor, has accumulated SVs and suggest that these modifications are a major driving force in the domestication process.

In the future, comprehensive evolutionary studies need to resequence an extensive number of genomes including wild relatives of crops. Unfortunately, a part of genetic diversity included in the wild grapevine lineages may be lost due to climate changes that occurred in the past or the anthropogenic pressures in the recent times (Figure 4). Accessions which originate from different geographical regions or that show different phenotypes, should be collected to maximise the diversity. Moreover, to acquire full knowledge of genetic diversity and to gain full understanding of genomic variations, the pan-genome of grapes should be planned. The concept is based on the investigation of the genetic variations by sequencing multiple individuals of a specific clade of evolution [141]. In the last few years, it has been widely accepted that the use of few reference genomes in evolutionary studies is limiting [137,142]. Although in grapevine the impact of domestication and breeding were weak, several resistance or tolerance genes could be lost, thus a pan-genomic study offers the possibility to recover and collect the gene diversity distributed in wild and domesticated forms. It may also be an important resource to explore the genomic architecture of grapevine as well as to estimate the evolution of lineages. The pan-genome analysis is applied recently for other crops and several are the reasons to export the same approach in grapevine [143]. Using the pan-genome information Gao et al. [144] have conducted a comparative analysis of cultivated tomatoes and close wild relatives, identifying genes selected or lost during domestication. Moreover, Sun et al. [145] showed that the phylogeny of the apple accessions inferred using a presence/absence variation pattern of pan-genomes was consistent with the phylogeny obtained by SNPs. Thus, genes involved in the domestication process can be mapped on the phylogenetic (including chronograms) and phylogeographic trees to understand as the genomes are changed through time by human activity and identify how many times and where the domestication occurred. Although we are conscious that the single-origin model is the most parsimonious hypothesis, other models can be tested and unexplored secondary domesticated processes or alternative introgression events could be observed. Moreover, we underline that demographic analyses are considered today the main approach to explore the origin of populations and can help to distinguish between possible centres of origin and diversification. Inferences in changes of population sizes throughout time applied to reconstruct the demographic history need several high-quality genome references, as well as the phase of the genotypes can be affected by genome reference biases [143].

Generally, the pan-genome refers to a full complement of genes of a species, but in grapes the concept could be expanded to the clade that includes the wild species of *Vitis* genus. Recent studies propose to produce a super-pan-genome for each crop, using at least one de novo assembly from each species [20]. In this way, it is possible to reduce the bias that could be produced mapping the sequencing data from genetically distant species. The *Vitis* genus counts around 60–70 taxa widespread throughout Eurasia and Northern America and several of them are inter-fertile [81,146]. Wild grapes are adapted to a wide range of climatic conditions and harbour genes resistant or tolerant towards both biotic and abiotic stresses, and several taxa are used today, as in the past, to produce rootstocks resistant to pathogens and pests, to drought and salinity, as well as cultivars characterised by a good quality of fruit and suitable for winemaking [147,148,149,150,151]. Thus, the genetic material conserved in wild grapes is a source of resistance to several stresses useful to improve the cultivars by traditional agronomic strategies or modern genomic editing methodologies [152]. We highlight the fact that the constitution of a super-pangenome could be an unprecedented resource to understand how to design the ideal cultivar of the future.

## 9. Concluding Remarks and Perspectives

Over the last two decades, an increasing number of studies aimed to examine the genetic resources of grapevine and to investigate its ancestries. In this review, we have discussed the main findings focused on answering important issues regarding the origin of the grapevine. Several studies proposed that the main domestication occurred in the southern regions of Caucasus, but recent insight suggests that a primo-domestication centre could be extended to Central Asian countries. In these areas, the grapevine has suffered from a history of protracted cultivation, thus, the pre-domestication phase may have begun earlier than previously thought. Even if these results fit well with the protracted model of crop domestication proposed by Allaby et al. [89], successive studies are needed to define the times of transition between different domestication phases. Grapevine cultivars were introduced to the Mediterranean basin only successively, but secondary centres of domestication and diversification are also highly debated. Several years of breeding seem to have obfuscated the signatures of the secondary domestication processes and recent molecular studies have shown that the role of introgression appears to have been fundamental. Introgression from *sylvestris* to *vinifera* aided by human activity have contributed to the domestication processes. Wild populations have had enough time to adapt to local environments and they harbour desirable traits useful for the improvement of the varieties. On the other hand, gene flow from *vinifera* to *sylvestirs* has been widely detected and it can have a significant impact on the conservation of wild populations. On the basis of data reported, the introgression is far from uncommon, thus, we propose that a detailed screening of gene flow which occurred between *sylvestris* and *vinifera* germplasms should be conducted urgently.

Today the grapevine domestication process is a challenge which is far from being completely solved and, though some aspects have found a preliminary answer, some issues remain pending: (1) routes of migration and secondary domestication centres should be verified by specific tests; (2) phylogeography of wild grapevine populations is understood but little is known about time and mode of diffusion in the Mediterranean basin; (3) new tools of evolutionary genomics should be applied to distinguish between domestication and diversification centres.

We argue that the realisation of a pan-genome of grapes could be a useful resource to track the change of genes which have occurred during different phases of domestication. In the future, whole-genome resequencing analysis will make it possible to explore a large portion of the variabilities in grapevine and some of the abovementioned pending issues are expected to be solved. However, we are also aware that only by increasing the sampling of landraces and wild populations distributed in remote regions and particularly in Central Asian countries, will it be possible to increase the chance to obtain a complete picture of genetic relationships and go back to the roots of the domestication process.

## Figures and Tables

**Figure 1 ijms-22-04518-f001:**
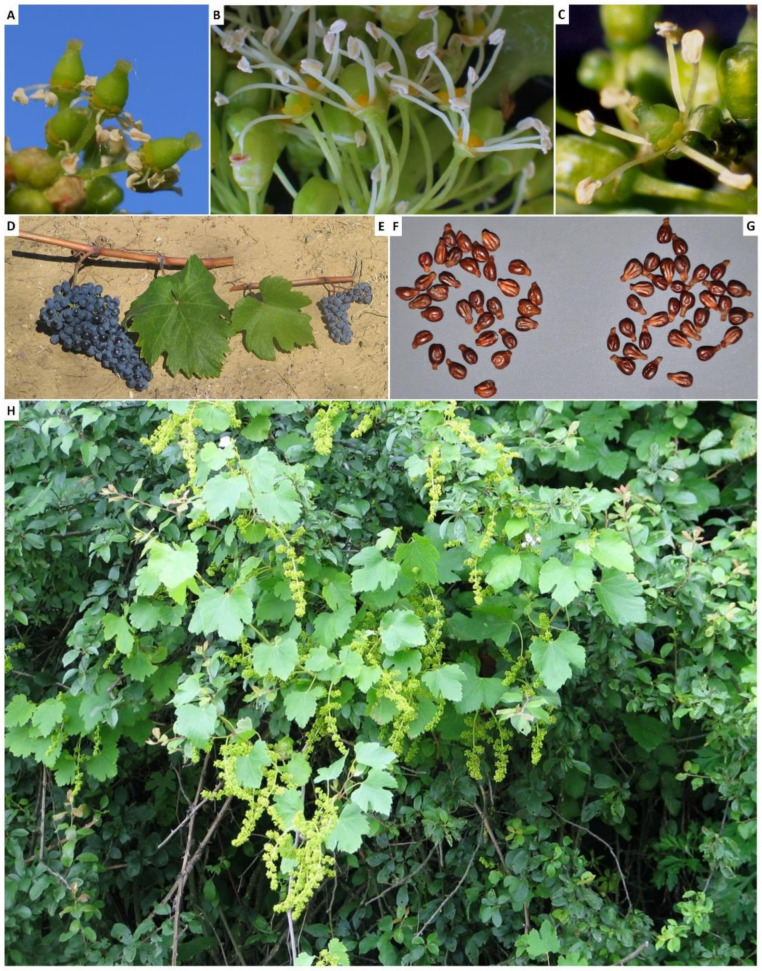
Phenotypical traits of *sylvestris* and *vinifera*. Female (**A**) and male (**B**) flowers of *sylvestris* individuals. Hermaphrodite flower (**C**) of *vinifera* individuals. Leaf and bunch of *vinifera* (**D**) and *sylvestris* (**E**) individuals. Seed of *sylvestris* (**F**) and *vinifera* (**G**) individuals. (H) Plant of *sylvestris* individual. Seeds of *sylvestris* are small, with a rounded outline and short stalks, while the *vinifera* ones are large, elongated, pyriform with elongated stalk. Leaves of *sylvestris* are smaller than the cultivated ones. *Vinifera* shows berries and bunches bigger than those of *sylvestris*, with a higher sugar content.

**Figure 2 ijms-22-04518-f002:**
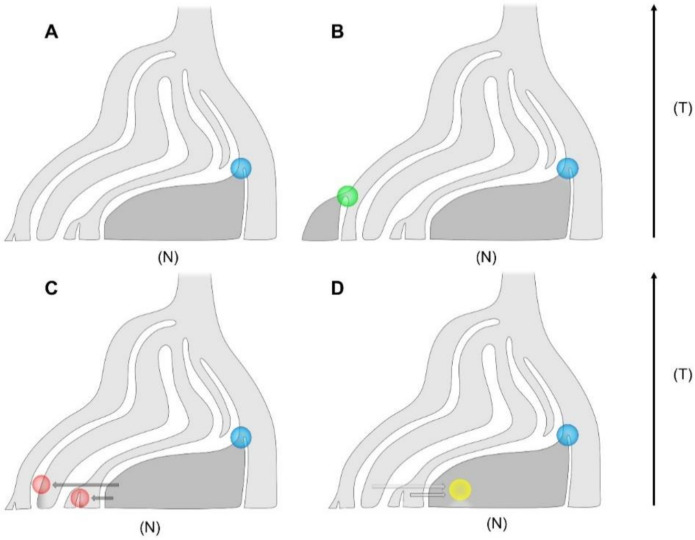
Main models of domestication proposed in the literature to explain the evolution of ancestral wild (light grey) and domesticated grapevine (dark grey). (**A**) Single-origin model; blue circle indicates a demographic bottleneck which occurred during the evolution of grapevine. (**B**) Multi-origin model; green circle indicates a secondary domestication event. (**C**) Multiple events of introgression from domesticated to wild grapevine (red circles) caused by gene flow (dark grey arrows). (**D**) Multiple events of introgression from wild to domesticated grapevine (yellow circle) caused by gene flow (light grey arrows). T = time (arrowhead indicates the past) and N = size of population.

**Figure 3 ijms-22-04518-f003:**
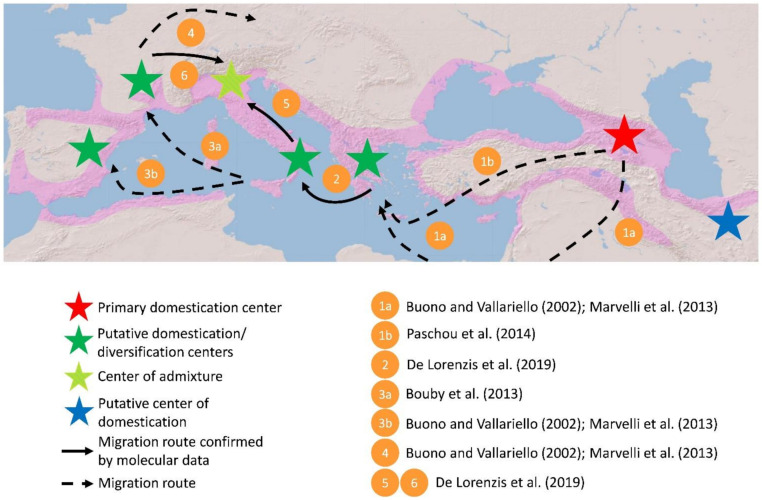
Map depicting probable grapevine domestication and diversification centres (stars) and the main grapevine migration routes (arrows). Pink area shows the distribution range of wild grapevine. According to many researchers, the domestication of grapevine took place around Mount Ararat in the Caucasus (red star). Its diffusion around the Mediterranean basin could have followed three main pathways. The first pathway goes from Mount Ararat to Mesopotamia, Egypt and Greece, considered a secondary domestication centre (green star), in the Bronze Age (arrow 1a). Others agree that grapevine arrived in Greece through Anatolia (arrow 1b). The second route starts from Greece and goes to *Magna Graecia* (Sicily, Southern Italy) (green star), France (Marseille) (green star) and Spain (green star) under the influence of the Greeks, Etruscans and Phoenicians (arrows 2, 3a and 3b). The third route goes from France to the north of Europe, mainly through the Rhone, the Rhine and the Danube, under the influence of the Roman Empire (arrow 4). Recently, Northern Italy (striped star) has been highlighted as an admixed centre of the Southern Italian (arrow 5) and Central European (arrow 6) population.

**Figure 4 ijms-22-04518-f004:**
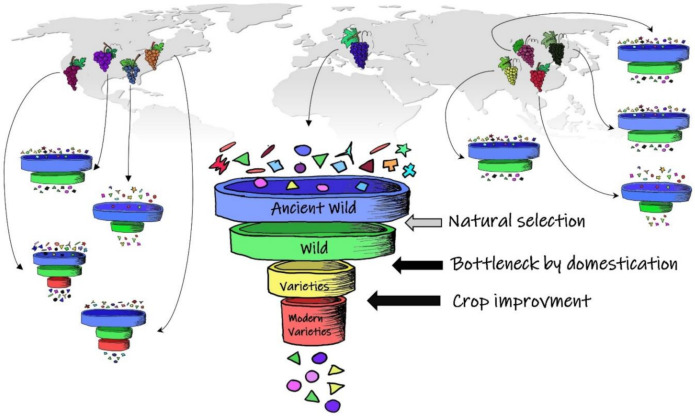
Grapevine is originated from an ancient wild progenitor. Although it has suffered from a weak bottleneck and the breeding have maintained a high level of heterozygosity, several resistance or tolerance genes could be lost during the evolution. The funnel shows the reduction of genetic diversity in time (different shapes indicate different classes of genes and different colours indicate different alleles). Natural selection (horizontal grey arrow) caused by climate changes in the past and artificial selection (horizontal black arrows) caused by human activity more recently can have driven the loss of genes and alleles. On the other hand, the wild American and Asian grapes are adapted to a wide range of climatic conditions and have suffered minor effects from human activity, thus they can harbour several resistance or tolerance genes. The assembly of a pangenome offers the possibility to recover and collect the original gene diversity conserved in wild and domesticated grapes.

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
