# Peer review of "Back to the Origins: Background and Perspectives of Grapevine Domestication"

_ijms, 2021, doi:10.3390/ijms22094518_

Round 1

Reviewer 1 Report

In general well written, clear and short, I have some minor edits/comments 

  • In Figure 1, panels G and H are not described in caption.
  • In Line 54 page 2, I think paragraph “To date, the…” should be part of the text not figure caption
  • Reference 79, pag. 16 line 747, author list is incomplete, please add et al., at the end of last author listed or add the other two authors.

Author Response

In general well written, clear and short, I have some minor edits/comments 

In Figure 1, panels G and H are not described in caption.

In Line 54 page 2, I think paragraph “To date, the…” should be part of the text not figure caption

Authors: We agree with the reviewer and we thank for the suggestion. We think that it should be a formatting error because it is not present in original manuscript. We have modified the text and the caption of figure1.

Reference 79, pag. 16 line 747, author list is incomplete, please add et al., at the end of last author listed or add the other two authors.

Authors: We have corrected the citation.

Reviewer 2 Report

The review submitted by Grassi and De Lorenzis deeply collect the findings of the last two decades on grape domestication and diversification. The overall manuscript is very well written and the topics are discussed with the support of the appropriate references. English language is used properly. The text does not need any major revision, just few suggestions.

Introduction. This section is too narrative. Focus on the differences between sylvestris and vinifera going straight to the points of the review. Merge the introduction with the following paragraph underlining pros and cons about domestication studies, what is missing and what questions are still unaswered. You can propose some questions, that will be answered in the review.

Check the legend of Figure 1, there are some typos. Part of the text is in the legend of the Figure and the G and H sections are not described. Please change sativa in vinifera in the legend.

Paragraph 2. The review is on grapevine, but with this paragraph the aim of the review is getting confused. Please merge it with the introduction, eliminating unusefull parts and storing the key messages (such as wild resources importance and feralization).

Paragraph 8. This section must be stronger. I suggest to hypothesize one or more experimental designs using the pangenomics approach, that you suggest, able to answer the doubts currently unsolved. For example, among the domestication models arising from the literature (Figure 1), what kind of experimental design could the scientific community consider to confirm one of the models?

Minor revisions:

Delete "many times" line 20

Delete "the" line 148

Change the step in lines 344-345 in "Indeed, different estimation methods can produce different results"

Delete "evolutionary" line 356

Delete the step in line 365 "The admixture...grapevine domestication"

Rephrase lines 419-423

Author Response

The review submitted by Grassi and De Lorenzis deeply collect the findings of the last two decades on grape domestication and diversification. The overall manuscript is very well written and the topics are discussed with the support of the appropriate references. English language is used properly. The text does not need any major revision, just few suggestions.

Introduction. This section is too narrative. Focus on the differences between sylvestris and vinifera going straight to the points of the review. Merge the introduction with the following paragraph underlining pros and cons about domestication studies, what is missing and what questions are still unaswered. You can propose some questions, that will be answered in the review.

Authors: We thank the reviewer for the important suggestions. We have discussed at length this issue, also with other colleagues. In a first version of manuscript, paragraphs 1 and 2 were merged, but for greater clarity, we have preferred to split the text in two paragraphs. The introduction was too long and dispersive, thus, after several efforts, we are convinced that the split in the two paragraphs is the clearer option. In paragraph 2 are described the main problems about the domestication, the importance of wild crop relatives and several information useful to understand the successive paragraphs. For our experience, the information reported in paragraph 2 is useful for readers not familiar with domestication. Moreover, we think that the title of successive paragraphs are sufficient to evoke the issues that we desire to explain in the text.  We consider that to add a list of questions still unanswered is redundant and, however, a list of unanswered issues is reported in the last paragraph. Moreover, we have followed the suggestion of the reviewer and we have added some information about Vitis vinifera in Introduction (lines 38-43).

Check the legend of Figure 1, there are some typos. Part of the text is in the legend of the Figure and the G and H sections are not described. Please change sativa in vinifera in the legend.

Authors: We think that it should be a formatting error because it is not present in the original manuscript. We agree with the reviewer and we have modified the text and the caption of the figure1.

Paragraph 2. The review is on grapevine, but with this paragraph the aim of the review is getting confused. Please merge it with the introduction, eliminating unusefull parts and storing the key messages (such as wild resources importance and feralization).

Authors: We have answered above, in Introduction.

Paragraph 8. This section must be stronger. I suggest to hypothesize one or more experimental designs using the pangenomics approach, that you suggest, able to answer the doubts currently unsolved. For example, among the domestication models arising from the literature (Figure 1), what kind of experimental design could the scientific community consider to confirm one of the models?

Authors: we have strengthened paragraph 8 as suggested by the reviewer. We have described how the information obtained by the pan-genome can help to explain the domestication process (lines 518-536)

Minor revisions:

Delete "many times" line 20

Authors: we cannot delete “many times” otherwise the meaning of the sentence is incorrect.

Delete "the" line 148 

Authors: we deleted the word (line 159).

Change the step in lines 344-345 in "Indeed, different estimation methods can produce different results" 

Authors: we reworded the sentence(see line 360).

Delete "evolutionary" line 356. 

Authors: we deleted the word (line 371).

Delete the step in line 365 "The admixture...grapevine domestication" 

Authors: we deleted the sentence (see line 380).

Rephrase lines 419-423

Authors: we reworded the sentences (lines 434-437).

Reviewer 3 Report

Review

Back to the origins: background and perspectives of grapevine domestication

By Fabrizio Grassi and Gabriella De Lorenzis

The vine genome is a useful resource to discover changes, which have occurred in the genomes and to improve our understanding about the domestication. The paper reviews results obtained in the main phylogenetic and genomic studies in the last two decades, to clarify the fundamental questions regarding where, when, and how many times grapevine domestication took place. Overall the paper is well written and clear. It addresses different aspects of the topic ranging from the question how many times domestication took place, what the plastid or the whole genome DNA tells us, reviews novel results using ancient DNA or seed shapes.

I have not substantial critizism, it was a pleasure to read the paper. My points below are facultative.

Major points:

  1. Chapter 3: Might the authors want to add 2-3 sentences to explain how secondary, teriary… etc. domestication can be distinguished from ‘simple’ genetic shift/breeding, e.g. by genetic components of wild vine species. Are they different (and how) at the different loci of domestication? In principle the answer is given at page 6, but a few words for reader not familiar with the details would improve the text.
  2. Chapter 5 and 6: There are certain uncertainties regarding when, where and how domestication occurs. Might the authors want to add a paragraph and compare the options of plasmid, microsatellite, SNP (arrays) and WGS studies, how development proceeds in the last years and what to expect? Partly an answer is given in Conclusion section (pangenome approach, more sampling…). I have in mind a more methods-centered opinion.

Minor points:

  1. The time arrow seems to point into wrong direction in Fig. 2

Author Response

The vine genome is a useful resource to discover changes, which have occurred in the genomes and to improve our understanding about the domestication. The paper reviews results obtained in the main phylogenetic and genomic studies in the last two decades, to clarify the fundamental questions regarding where, when, and how many times grapevine domestication took place. Overall the paper is well written and clear. It addresses different aspects of the topic ranging from the question how many times domestication took place, what the plastid or the whole genome DNA tells us, reviews novel results using ancient DNA or seed shapes.

I have not substantial critizism, it was a pleasure to read the paper. My points below are facultative.

Major points:

  1. Chapter 3: Might the authors want to add 2-3 sentences to explain how secondary, teriary… etc. domestication can be distinguished from ‘simple’ genetic shift/breeding, e.g. by genetic components of wild vine species. Are they different (and how) at the different loci of domestication? In principle the answer is given at page 6, but a few words for reader not familiar with the details would improve the text. 
  2. Chapter 5 and 6: There are certain uncertainties regarding when, where and how domestication occurs. Might the authors want to add a paragraph and compare the options of plasmid, microsatellite, SNP (arrays) and WGS studies, how development proceeds in the last years and what to expect? Partly an answer is given in Conclusion section (pangenome approach, more sampling…). I have in mind a more methods-centered opinion.

Authors: We thank the reviewer for the important suggestions and we answer both issues. Since the comparison among different methods is widely described in paragraph 3, as well as the different models of domestication (secondary, teriary… etc), we have preferred to avoid adding redundant information. On the other hand, we recognize that paragraph 8 can be strengthened further and we have added some sentences to explain how the pangenome analysis can help to answer the doubts currently unsolved about where and how many times the domestication occurred (lines 518-536). 

Minor points:

  1. The time arrow seems to point into wrong direction in Fig. 2 

Authors: for a better understanding of the figure, we changed the caption (line 180). We have added the word: arrowhead